# Collaborative Metapath Enhanced Corporate Default Risk Assessment on Heterogeneous Graph

Anonymous Author(s)*†

## ABSTRACT

Default risk assessment for small companies is a tough problem in financial services. Most small businesses expose fragility to external impacts and few information on their insecure finances. Recent efforts utilize advanced Heterogeneous Graph Neural Networks (HGNNs) with metapaths to exploit interactive features in corporate activities for risk analysis. However, few works are proposed for traditional commercial banks. Given a real financial graph, how to detect corporate default risks? We identify two challenges for the task. (1) Massive noisy connections hinder HGNNs to achieve strong results. (2) Multiple semantic connections greatly increase transitive default risk, while existing hierarchical aggregation schemes do not leverage such connection patterns.

In this work, we propose a novel Heterogeneous Graph Co-Attention Network (HetCAN) for corporate default risk assessment. HetCAN aims to take advantage of collaborative metapaths to distill effective risky features by a co-attentive aggregation mechanism, consisting of two attention scores and pairwise importance learning. First, the local attention score models the importance of neighbors under each metapath by considering holistic metapath context. Second, the global attention score further adjusts the importance of neighbors by combining these local attention scores to filter valuable/noisy signals. Then, HetCAN employs pairwise importance learning to enhance attention scores of multi-metapath neighbors for risky feature distillation. Extensive experiments verify that HetCAN outperforms state-of-the-art methods in accurately predicting default risks on large-scale banking datasets.

## CCS CONCEPTS

• **Information systems** → **Data mining**; • **Applied computing** → **Enterprise modeling**; • **Computing methodologies** → **Neural networks**.

## KEYWORDS

default risk, finance, heterogeneous graph, graph neural network, attention mechanism

**ACM Reference Format:**
Anonymous Author(s). 2023. Collaborative Metapath Enhanced Corporate Default Risk Assessment on Heterogeneous Graph. In *Proceedings of ACM Conference (Conference'17)*. ACM, New York, NY, USA, 11 pages. https://doi.org/10.1145/nnnnnnn.nnnnnnn

## 1 INTRODUCTION

Corporate default risk assessment is a fundamental problem, that lies at the cornerstone of various financial services. In particular, early risk warning for small companies is crucial to prevent considerable losses. Unlike the large enterprises, most small businesses are fragile to external impacts and their financial stability is considered weak. Here, our task is to predict whether a small company will fail to repay its loans in the future. Conventional methods [7, 36, 50] infer the default probability by machine learning models based on large amounts of historical data. However, since small companies scarcely publish financial reports and statements, the modeling usually suffers from deficient and outdated data. Furthermore, learning-based methods treat each company solely, but seldom fully exploit interactions rich in financial activities, which leads to unsatisfactory prediction results [2, 48, 54].

A modern company constantly leaves traces in the digital world. For example, the company transfers money to its counterparty in a transaction system. Thus, each company obtains a large number of connections, which naturally form a graph with abundant semantics. In fact, such graph contains valuable signals for risk assessment. To take advantage of interactive data, we adopt heterogeneous graphs (HGs) for modelling the problem. HGs are a powerful tool to represent real-world systems through a series of objects (nodes) and relations (edges) with diverse types [37]. Figure 1a illustrates an example of HG schema, which contains three views: fund view denotes transaction relations, industry view denotes industrial chain relations, and equity view denotes investment relations. Additionally, we adopt metapath [38] (ordered sequence of node and edge types) to model specific semantic connections between companies in our scenario, as shown in Figure 1b.

To accurately spot default risks, we conduct an in-depth exploration on real financial graphs and obtain two observations. First, a few semantic connections preserve information strongly related to default risks, while most are noisy and irrelevant. For example, default companies universally look low-risk, mainly by their daily money transfer links with normal hubs (like public utilities, online platforms). Second, multiple semantic connections increase the transitive default risk. Concretely, the connection patterns where companies are linked by diverse metapaths have great impacts on default prediction. It is motivated to extract the risky factors from such locally connected structures, called "*transitive risky features*". Detailed analyses are presented in Section 2.2 to uncover the key graph properties toward our assessment task.

Given a HG under such setting, corporate default risk assessment can be further regarded as a node classification problem. Recently, deep learning on HGs has experienced great advancements, with Heterogeneous Graph Neural Networks (HGNNs) showing strong results on a wide range of tasks: recommendation [9, 10, 47, 53], fraud detection [20, 27, 30, 45], traffic forecast [15, 33] and financial analysis [17, 28, 42, 55]. HGNNs derive node representations

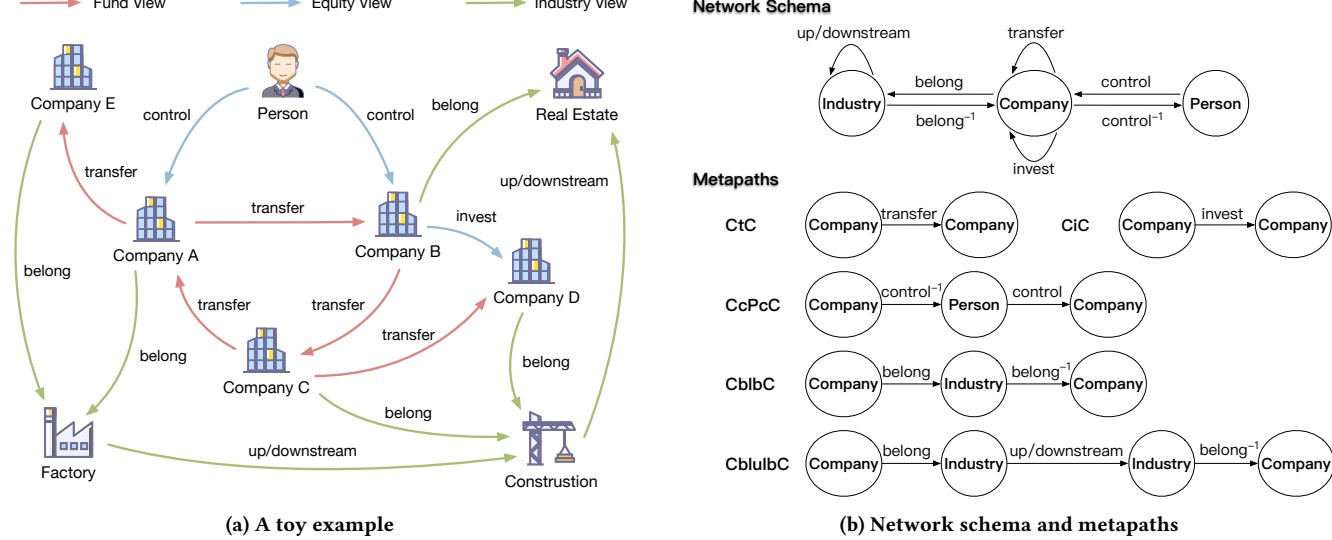

**(a) A toy example**

**(b) Network schema and metapaths**

Figure 1: The illustrative example for modeling the scenario via heterogeneous graph.

mainly based on hierarchical aggregation, which adopt metapaths to capture neighborhood information of the same semantic and then fuse diverse semantics.

However, based on the above observations, HGNNs have to face two challenges. (1) Massive noisy connections hinder HGNNs to obtain satisfying results. The general message passing aggregates metapath-based neighbors regardless of whether they are similar to the target node. A few works [20, 51, 52] attempt to improve the aggregation by a class of similarity metrics against noises, but they discard intermediate nodes and edges along metapath, which are beneficial to noise discrimination. (2) Hierarchical aggregation schemes do not fully exploit multiple semantic connections. Typical HGNNs utilize each metapath separately in fine-grained aggregation and disregard the correlation among diverse metapaths [41]. Intuitively, such connection patterns indicate higher importance and correlation between companies, which contribute to learning the transitive risky features. Thus, existing methods incur information loss and easily lead to suboptimal prediction.

In this paper, we propose a novel **Het**erogeneous Graph **Co-A**ttention **N**etwork (HetCAN) for corporate default risk assessment. Our model aims to make use of collaborative metapaths to learn effective risky features by a well-designed co-attention mechanism. Specifically, we introduce two attention scores. First, the local attention score models the similarity-based importance of neighbors to the target node by considering holistic metapath context, which makes full use of node and edge information along metapath. For this purpose, we devise a metapath context encoder by recurrent skipping networks. Second, the global attention score adjusts the importance of neighbors by fusing local attention scores under different metapaths to discern valuable/noisy signals. To further exploit multiple semantic connections, we propose a pairwise importance learning method, which divides global attention scores into two groups with respect to the number of metapaths, and optimizes

a margin loss to increase the importance weights of multi-metapath neighbors, guiding the distillation of transitive risky features.

Our main contributions are summarized as follows:

- We conduct quantitative analyses on real financial graphs to confirm the intuition that multiple semantic connections have great impact on the small company defaults, while in the presence of massive noisy connections.
- We propose a novel HetCAN for corporate default risk assessment on HGs. The model carefully takes advantage of collaborative metapaths to characterize risky features by a co-attention mechanism, consisting of two attention scores and pairwise importance learning.
- We thoroughly evaluate the proposed model on large-scale banking datasets with 24.81 million nodes and 129.54 million edges. The results demonstrate that HetCAN outperforms other competitors in predicting default risks for small companies. The code has been open-sourced[1].

## 2 PRELIMINARY

In this section, we present the problem statement of our work and the empirical observations of real financial graph.

### 2.1 Problem Statement

Company default is not only driven by intrinsic behaviors but derived from externally associated entities as well. It turns out that financial institutions have abundant interactive relations between companies (see Figure 1a). Thus, our focus is to make use of interactive data for corporate default risk assessment.

**Definition 2.1** *Heterogeneous Graph*. A heterogeneous graph is denoted as $\mathcal{G} = (\mathcal{V}, \mathcal{E})$, where $\mathcal{V}$ is a node set associated with a

---

[1]https://github.com/adlington/HetCAN

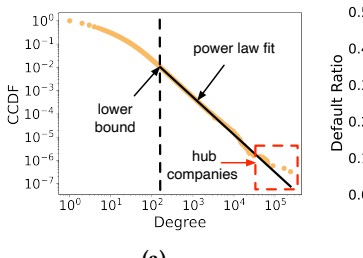
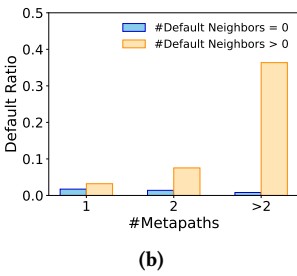

(a)

(b)

**Figure 2: Connection analysis. (a) Empirical degree distribution of fund view and its power law fit in a log-log scale, where the vertical axis represents the complementary cumulative distribution function (CCDF). (b) The default ratio under different number of metapaths.**

node type mapping function $\psi: \mathcal{V} \rightarrow \mathcal{A}$, and $\mathcal{E}$ is an edge set associated with an edge type mapping function $\vartheta: \mathcal{E} \rightarrow \mathcal{R}$. Respectively, $\mathcal{A}$ and $\mathcal{R}$ are the sets of node and edge types, with $|\mathcal{A}| + |\mathcal{R}| > 2$.

Figure 1b shows the graph schema of $\mathcal{G}$, which contains three types of nodes and five types of edges with respective attributes. Formally, for a node type $A \in \mathcal{A}$, $\mathbf{X}_A \in \mathbb{R}^{|\mathcal{V}_A| \times d_A}$ denotes the node attribute matrix with $|\mathcal{V}_A|$ nodes and $d_A$ attributes. Likewise, for an edge type $R \in \mathcal{R}$, $\mathbf{X}_R \in \mathbb{R}^{|\mathcal{E}_R| \times d_R}$ denotes the edge attribute matrix with $|\mathcal{E}_R|$ edges and $d_R$ attributes.

**Definition 2.2 Metapath.** A metapath $\Phi$ is a path defined in the form of $A_1 \xrightarrow{R_1} A_2 \xrightarrow{R_2} \cdots \xrightarrow{R_{l-1}} A_l$, which describes a collection of relations $R_1 \circ R_2 \circ \cdots \circ R_{l-1}$ between $A_1$ and $A_l$, where $\circ$ is the composition operator on relations.

Given a metapath $\Phi$ of a HG, a **metapath instance** $\phi$ is a node sequence following the schema defined by $\Phi$. The **metapath-based neighbors** $\mathcal{N}_v^\Phi$ are the nodes which connect to node $v$ via metapath instances of $\Phi$. The **metapath context** $\mathcal{H}_{uv}^\Phi$ is the set of metapath instances of $\Phi$ between node $u$ and $v$. Besides, our approach can jointly use node and edge information along metapath, which we refer to as "*holistic metapath context*".

The formalization of our problem is as follows.

**Problem 2.1 Corporate Default Risk Assessment.** Our purpose is to infer whether a small company will be default over a period of time. In particular, given the heterogeneous graph $\mathcal{G} = (\mathcal{V}, \mathcal{E})$ and the training set $\mathcal{D} = \{(v, y_v)\}$ (where $y_v \in \{0, 1\}$, i.e. normal/default), the task is to predict the default probability $\hat{y}_u$ of each small company $u \in \mathcal{U}$ (where the testing set $\mathcal{U} \subseteq \mathcal{V}$), based on both features and interactive information distilled from $\mathcal{G}$.

## 2.2 Exploratory Analysis

We empirically demonstrate that the key graph properties have obvious impacts to corporate default risk assessment. The statistical results are reported in Figure 2, which are conducted on the real financial graph data (see Section 4.1). The observations inspire us to take advantage of these graph properties for modeling, as detailed below (see Appendix A.1 for other analyses).

**Noisy connection analysis.** We investigate fund view (i.e. transaction relations) to demonstrate the prevalent noisy connections. In general, financial networks conform to the scale-free properties [12, 19]. Thus, we fit the degree distribution of fund view to a power-law distribution by using a released statistical method [6]. Figure 2a exhibits the power-law behavior with lower bound around 160 and exponent around 2.6, where the large-degree nodes (i.e. hub companies) shown in the bottom right corner have a fatter tail than the fitted distribution. We further study these hub companies from three perspectives: 1) Topology: approximately 0.004% of all the companies generate transaction connections with at least 27.3% companies. 2) Industry: a majority of them are public service companies, engaged in the fields of tax agency, electric power, insurance, etc. 3) Behaviors: compared to the overall statistics, the average transaction amount decreases by 37.8% and the average transaction volume increases by 63.1%. Inspired by these behaviors, we leverage edge information to identify noisy connections.

**Multiple semantic connection analysis.** We next study the correlation between connection patterns and default risks. First, we collect the metapath-based neighbors of each small company. Then, we divide companies into separate groups: companies with default neighbors and companies with no default neighbors. We further count the number of metapaths between companies and neighbors for comparison. Figure 2b shows the results, where the default ratio (i.e. the proportion of default companies) is calculated in each group respectively. We observe that companies with default neighbors have higher default risk than those without default neighbors. Additionally, more metapaths between companies and default neighbors lead to higher default risk. Thus, it is promising to characterize the transitive risky features from neighbors with multiple semantic connections for the problem.

## 3 METHODOLOGY

In this section, we introduce the proposed HetCAN. Figure 3 shows the overall framework, which is divided into four parts: (a) Heterogeneous content encoding distills holistic metapath context in a hierarchical manner. (b) Co-attentive aggregation adopts two attention scores, including local attention score to model the similarity-based importance of neighbors and global attention score to further adjust the importance of neighbors, yielding node-level attention values. (c) Pairwise importance learning exploits multiple semantic connections to extract transitive risky features, by optimizing a proposed margin loss. (d) Semantic fusion learns the importance over diverse metapaths, yielding final node embeddings for the downstream task. The algorithm is stated in Appendix A.2.

## 3.1 Heterogeneous Content Encoding

We begin with the heterogeneous content encoding to exploit aspects of information from metapaths. Existing HGNNs [11, 44] discard intermediate nodes and edges along metapath, which results in significant loss of information. Hence, we design a metapath context encoder to distill the semantic information ingrained in metapath instances by using recurrent skipping networks (RSNs) [13]. By integrating RNNs with residual learning, RSNs capture the relational dependencies of nodes, which focus on learning from paths for knowledge graph (KG) embedding.

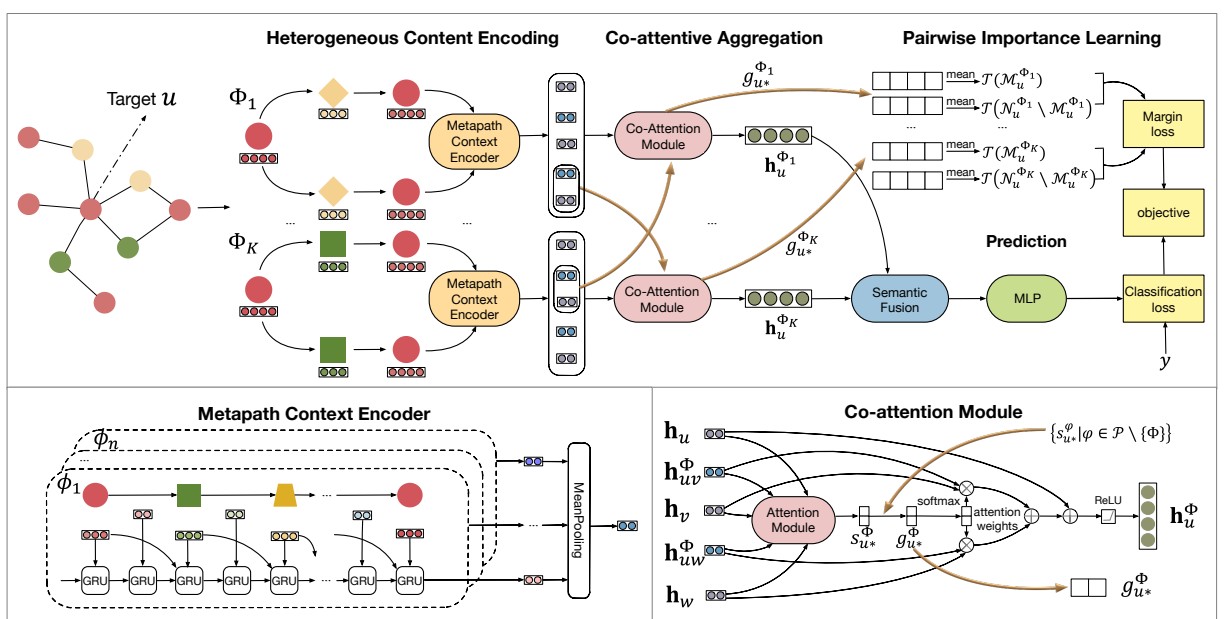

**Figure 3: The architecture of the proposed model.**

*3.1.1 Feature transformation.* We first project feature vectors into a unified latent vector space. Different node types and edge types have separate feature spaces, which cause obstacles for exploiting neighborhood information. Thus, we represent each type of nodes and edges via a simple linear transformation. Given a node type $A \in \mathcal{A}$ and an edge type $R \in \mathcal{R}$, for each node $v \in \mathcal{V}_A$ and each edge $e \in \mathcal{E}_R$, we define the projection as below:

$$\mathbf{h}_v = \mathbf{W}_A \mathbf{x}_v, \quad \mathbf{h}_e = \mathbf{W}_R \mathbf{x}_e \tag{1}$$

where $\mathbf{x}_v \in \mathbf{X}_A$ and $\mathbf{x}_e \in \mathbf{X}_R$ are the initial attribute vectors, $\mathbf{h}_v \in \mathbb{R}^d$ and $\mathbf{h}_e \in \mathbb{R}^d$ are the projected embedding vectors of nodes and edges, $\mathbf{W}_A \in \mathbb{R}^{d \times d_A}$ and $\mathbf{W}_R \in \mathbb{R}^{d \times d_R}$ are type-specific weight matrices. Let $d$ be the latent vector size.

*3.1.2 Metapath instance encoding.* We next use RSNs to encode all nodes and edges along metapath. Given a metapath instance $\phi = v_1 \xrightarrow{e_1} v_2 \xrightarrow{e_2} \cdots \xrightarrow{e_l} v_{l+1}$, we first define the embedding sequence of $\phi$, formed as $\{\mathbf{z}_1, \mathbf{z}_2, \ldots, \mathbf{z}_{2l+1}\}$, which satisfies: when $t$ is an odd number, $\mathbf{z}_t$ is the projected node embedding, i.e. $\mathbf{z}_t = \mathbf{h}_v$, where $v = v_i$ and $i = \frac{t+1}{2}$; otherwise, $\mathbf{z}_t$ is the projected edge embedding, i.e. $\mathbf{z}_t = \mathbf{h}_e$, where $e = e_j$ and $j = \frac{t}{2}$. Then, we propose using the skipping operation of the following form:

$$\mathbf{h}'_t = \begin{cases} \text{GRU}\left(\mathbf{h}'_{t-1}, \mathbf{z}_t\right) & t = 1, 3, \ldots, 2l+1 \\ \mathbf{W}_H \text{GRU}\left(h'_{t-1}, z_t\right) + \mathbf{W}_V \mathbf{z}_{t-1} & t = 2, 4, \ldots, 2l \end{cases} \tag{2}$$

where $\mathbf{h}'_t$ is the output hidden state of RSN at time step $t$, and $\mathbf{W}_H, \mathbf{W}_V \in \mathbb{R}^{d \times d}$ are parameter weight matrices shared at different time steps. We adopt GRU [5] as RNN units to learn long-term dependencies in the embedding sequence. The final output vector $\mathbf{h}'_{2l+1}$ is the distilled embedding of the given metapath instance, i.e. $\mathbf{h}_\phi = \mathbf{h}'_{2l+1}$.

*3.1.3 Metapath context encoding.* Finally, we apply the mean pooling operation to derive the embedding vector of metapath context, formed as

$$\mathbf{h}^\Phi_{uv} = \frac{1}{|\mathcal{H}^\Phi_{uv}|} \sum_{\phi \in \mathcal{H}^\Phi_{uv}} \mathbf{h}_\phi \tag{3}$$

where $\mathcal{H}^\Phi_{uv}$ is the metapath context between $u$ and $v$.

## 3.2 Co-attentive Aggregation

We next state how to take advantage of collaborative metapaths to learn effective risky features. HGNNs [11, 44, 49] often adopt hierarchical aggregation, including node-level and semantic-level attentions, to learn the importance of nodes and metapaths. The node-level attention is extended from classic GAT, which suffers from the limitation of discerning valuable/noisy signals. Although recent works [20, 51, 52] improve the aggregation by a class of node similarity metrics against noisy neighbors, it still remains two problems in face of real financial graphs. (1) The data deficiency of small companies impairs the performance of attention scoring methods based on node feature similarity. For example, some upstream and downstream companies have dissimilar features, while they have high-volume transactions, implying strong importance. (2) The attention scoring methods, which rely on single semantic connections, struggle to discern redundant information. Recall that the connection patterns where companies are linked with multiple types of metapaths can benefit the identification of transitive risky features (see Section 2.2).

To tackle these problems, we take advantage of more information (including holistic metapath context and collaborative metapaths) to filter valuable/noisy signals. A key idea of our approach is that we introduce the following two attention scores.

*3.2.1 Local attention score.* The local attention score $s_{uv}^\Phi$ models the similarity-based importance of neighbor $v$ to target node $u$ under a metapath $\Phi$. We estimate $s_{uv}^\Phi$ from two measurable aspects. First, we consider the similarity of node features. Intuitively, assume that neighbors with similar features are more important for the target node than dissimilar ones. Second, we consider the holistic metapath context for better importance estimation. Formally,

$$s_{uv}^\Phi = \sigma \left( \eta_\Phi \mathbf{h}_u \cdot \mathbf{h}_v + (1 - \eta_\Phi) \, \mathbf{a}_\Phi^\top \cdot \mathbf{h}_{uv}^\Phi \right) \qquad (4)$$

where $\sigma(\cdot)$ is the activation function and we choose $\tanh(\cdot)$ in this work, $\eta_\Phi$ is the metapath-specific learnable parameter, $\mathbf{h}_u \cdot \mathbf{h}_v$ is the dot-product similarity of node embeddings. The linear weighting operation $\mathbf{a}_\Phi^\top \cdot \mathbf{h}_{uv}^\Phi$ will take the holistic metapath context into calculation, where $\mathbf{a}_\Phi \in \mathbb{R}^d$ is the parameterized attention vector, $\mathbf{h}_{uv}^\Phi$ is the metapath context embedding.

*3.2.2 Global attention score.* We further distill the importance of metapath-based neighbor $v$ to target node $u$ by considering other metapaths. The choice is to fuse the local attention scores between $u$ and $v$ under different metapaths into the global attention score $g_{uv}^\Phi$, which is defined as follows:

$$g_{uv}^\Phi = (1 - \delta) s_{uv}^\Phi + \delta \sum_{\varphi \in \mathcal{P} \backslash \{\Phi\}} s_{uv}^\varphi \qquad (5)$$

where $\delta$ is the learnable parameter, $\mathcal{P}$ is the metapath set and $\mathcal{P} \backslash \{\Phi\}$ includes the rest metapaths. Let $s_{uv}^\varphi = 0$ if $v$ and $u$ are not connected via metapath $\varphi$. We use $\delta$ to adjust the weight for other metapaths. In particular, when $\delta = 0$, formula (5) returns to the local attention score. Note that we adopt $\delta$ for collaborative metapaths, avoiding the rise of model complexity (see Appendix A.3).

*3.2.3 Neighbor aggregation.* We normalize the global attention score with the softmax function, yielding the attention $\alpha_{uv}^\Phi$ as follows:

$$\alpha_{uv}^\Phi = \frac{\exp \left( g_{uv}^\Phi \right)}{\sum_{i \in \mathcal{N}_u^\Phi} \exp \left( g_{ui}^\Phi \right)} \qquad (6)$$

The final attention $\alpha_{uv}^\Phi$ is used to aggregate neighbor messages as

$$\mathbf{m}_u^\Phi = \mathbf{W}_M^\Phi \sum_{v \in \mathcal{N}_u^\Phi} \alpha_{uv}^\Phi \left[ \mathbf{h}_v \| \mathbf{h}_{uv}^\Phi \right] \qquad (7)$$

where $\mathbf{W}_M^\Phi \in \mathbb{R}^{d \times 2d}$ is the weight matrix, $\|$ is the vector concatenation.

Then, we adopt the pre-activation residual connections [26] for learning the metapath-specific embedding, which is to avoid over-smoothing. We have

$$\mathbf{h}_u^\Phi = \sigma \left( \mathbf{m}_u^\Phi + \mathbf{h}_u \right) \qquad (8)$$

where $\sigma(\cdot)$ is the ReLU activation function.

## 3.3 Pairwise Importance Learning

In this section, we introduce how to utilize multiple semantic connections to guide feature learning. Inspired by [39], we define the problem as learning to rank attention scores. The aim is to increase the importance of neighbors linked with multiple metapaths, ensuring higher attention over those with a single metapath.

For each node $u$, we first divide the metapath-based neighbors $\mathcal{N}_u^\Phi$ into two groups: multi-metapath neighbors and single-metapath neighbors, denoted by $\mathcal{M}_u^\Phi$ and $\mathcal{N}_u^\Phi \backslash \mathcal{M}_u^\Phi$. Then, we compute the average of global attention scores for each group:

$$\mathcal{T} \left( \mathcal{M}_u^\Phi \right) = \frac{1}{\left| \mathcal{M}_u^\Phi \right|} \sum_{v \in \mathcal{M}_u^\Phi} g_{uv}^\Phi \qquad (9)$$

$$\mathcal{T} \left( \mathcal{N}_u^\Phi \backslash \mathcal{M}_u^\Phi \right) = \frac{1}{\left| \mathcal{N}_u^\Phi \backslash \mathcal{M}_u^\Phi \right|} \sum_{v \in \mathcal{N}_u^\Phi \backslash \mathcal{M}_u^\Phi} g_{uv}^\Phi \qquad (10)$$

Intuitively, while $\mathcal{T} \left( \mathcal{M}_u^\Phi \right)$ is higher than $\mathcal{T} \left( \mathcal{N}_u^\Phi \backslash \mathcal{M}_u^\Phi \right)$, it is more likely to obtain higher importance weights of the multi-metapath neighbors. Hence, we propose a multi-metapath importance aware loss function, aiming to maximize the gap between the importance weights of multi-metapath neighbors and single-metapath neighbors. The loss function is defined as

$$\mathcal{L}_{margin} = \sum_{\Phi \in \mathcal{P}} \sum_{u \in \mathcal{V}} \max \left( 0, \rho - \left( \mathcal{T} \left( \mathcal{M}_u^\Phi \right) - \mathcal{T} \left( \mathcal{N}_u^\Phi \backslash \mathcal{M}_u^\Phi \right) \right) \right) \qquad (11)$$

where $\rho$ is a constant margin threshold.

Unlike pairwise margin losses [43, 46], we optimize the loss function by keeping distance between different groups above the certain threshold without narrowing distances between same groups. This is because neighbors under the same number of metapaths may have different importance, thereby encouraging them close may hinder effective feature learning. Additionally, we use average value instead of unit value for expressing the importance of each group in order to relax the constraints from traditional pairwise losses, accelerating convergence of the loss function.

## 3.4 Semantic Fusion and Model Learning

*3.4.1 Semantic fusion.* So far, we have obtained the metapath-specific embeddings of node $u$ in the form of $\left\{ \mathbf{h}_u^{\Phi_1}, \mathbf{h}_u^{\Phi_2} \ldots, \mathbf{h}_u^{\Phi_K} \right\}$. We next model the importance over diverse metapaths by semantic-level aggregation. The attention weights of metapaths are computed by the following operations.

$$\beta_{\Phi_i} = \frac{\exp \left( \mathbf{a}_S^\top \cdot \sigma \left( \mathbf{W}_S \mathbf{h}_u^{\Phi_i} \right) \right)}{\sum_{\Phi_j \in \mathcal{P}} \exp \left( \mathbf{a}_S^\top \cdot \sigma \left( \mathbf{W}_S \mathbf{h}_u^{\Phi_j} \right) \right)} \qquad (12)$$

$$\mathbf{q}_u = \sum_{\Phi_i \in \mathcal{P}} \beta_{\Phi_i} \mathbf{h}_u^{\Phi_i} \qquad (13)$$

where $\mathbf{a}_S \in \mathbb{R}^d$ is the learnable semantic preference vector, $\mathbf{W}_S \in \mathbb{R}^{d \times d}$ is the weight matrix, $\sigma(\cdot)$ is the activation function and here we use $\tanh(\cdot)$, and $\beta_{\Phi_i}$ is the attention weights over metapath $\Phi_i$. The output embedding $\mathbf{q}_u$ is summed by a weighted average of all the metapath specific embeddings of the target node $u$.

*3.4.2 Model learning.* We next feed the fused embeddings to MLP classifier for predicting the default probability, as follows:

$$\hat{y}_u = \text{MLP} \left( \mathbf{q}_u \right) \qquad (14)$$

The overall objective function of our proposed HetCAN model is a combination of margin loss and classification loss. Formally,

$$\mathcal{L} = \mathcal{L}_{clf} + \lambda \mathcal{L}_{margin} + \mathcal{L}_{reg} \qquad (15)$$

where $\lambda$ is the weight for the margin loss, $\mathcal{L}_{reg}$ is the L2 regularization to prevent over-fitting, $\mathcal{L}_{clf}$ is the classification loss of the following form:

$$\mathcal{L}_{clf} = -\frac{1}{|\mathcal{D}|} \sum_{(u, y_u) \in \mathcal{D}} y_u \log \hat{y}_u + (1 - y_u) \log (1 - \hat{y}_u) \quad (16)$$

## 4 EXPERIMENTS

In this section, we conduct extensive experiments on real-world datasets for evaluating the effects of our proposed methods.

### 4.1 Experimental Settings

*4.1.1 Datasets.* We evaluate HetCAN on two real-world datasets collected from a major commercial bank in China. In Table 1, SC21H1 contains about 252 thousand small companies (Jan.01, 2021 - Jun.30, 2021) for training and testing and SC21H2 contains about 247 thousand small companies (Jul.01, 2021 - Dec.31, 2021) for out-of-time testing. The size of training/validation/testing set of SC21H1 is set to 0.5/0.2/0.3. All the samples refer to small companies with no loans that obtained their loan by bank within the given time. We adopt the ground truth labels by financial experts. The positive samples are the small companies that have delayed or failed in the payments of principal and interests over seven days. Totally, we collect about 10 thousand default samples with the default ratio close to 2.0%.

We build HGs on the beginning time of each dataset, which consist of over 24.81 million nodes and 129.54 million edges (see Table 1). Note that over 95% of the company nodes are small companies. To enrich information for nodes, we exclude features with miss rate over 90% and extract 456 attributes for company node, from the aspects of corporate user profile, credit status, contractual capability, business information, and account behavior. For other types of nodes and edges, we collect a certain number of attributes ranging from 5 to 22 except the belong relation, such as personal credit score, industrial scale, amount, frequency, etc.

*4.1.2 Baselines.* We compare HetCAN against diverse baseline methods. Concretely, the traditional machine learning methods include Logistic Regression (LR), LightGBM [21] and Multi-Layer Perception (MLP), which only use target node features to train classifiers. The homogeneous GNNs include GCN [22], GraphSAGE [14] and GAT [40], which capture node features and structural information but overlook heterogeneity. The HGNNs include HAN [44], MAGNN [11], HGT [18] and Simple-HGN [34], which exploit heterogeneous node features and structural information.

We derive the following three variants of HetCAN.

- HetCAN$_{\backslash E}$ only considers node feature similarity in local attention score, i.e. without holistic metapath context.
- HetCAN$_{\backslash A}$ only adopts local attention score for attention calculation, i.e. without global attention score.
- HetCAN$_{\backslash L}$ removes the margin loss for training, i.e. without pairwise importance learning.

*4.1.3 Implementation and evaluation.* Our implementation is based on DGL with PyTorch backend. For the large datasets, we do neighbor sampling on HGs and the sampling size of neighbors is set to 10. We randomly initialize the model parameters with a Xavier

**Table 1: The statistical information of datasets.**

| Dataset | Node | Edge | #Label | Default Ratio |
|---|---|---|---|---|
| SC21H1 | Company:12,793,669 Person: 12,014,090 Industry: 447 | Transfer:102,605,728 Belong:11,349,549 Up/downstream:1,316 Control:13,832,259 Invest:1,754,576 | 251,963 | 1.96% |
| SC21H2 | Company:13,052,256 Person: 12,332,867 Industry: 447 | Transfer:103,478,322 Belong:11,973,522 Up/downstream:1,316 Control:14,854,227 Invest:1,777,744 | 247,165 | 2.09% |

initializer and choose Adam as the optimizer. Moreover, we set the batch size to 256, the learning rate to 0.001, the margin value to 1 and set the dropout rate to 0.5 and the weight decay to 0.01 to prevent overfitting. We also perform early stopping during the training if the validation performance is not improved for 30 epochs. The parameters of baselines are set up either as their default values or the same as in our model (see baseline settings in Appendix A.4). For homogeneous GNNs, we discard the node and edge type of HGs and use it as the input. Note that we will run each algorithm 5 times to report the average results.

In addition, the model performance is evaluated mainly via AUC (Area Under the ROC Curve) and KS (Kolmogorov Smirnov), which are extensively applied in banking scenarios. Higher scores of AUC and KS signify better prediction on defaults.

### 4.2 Real-world Performance

We evaluate different methods on real-world datasets. As shown in Table 2, the main observations are summarized as follows:

- Our HetCAN model outperforms other competitors by a significant margin on both datasets. Its AUC and KS improve at least 1.7% and 7.5% compared with the best performance of other methods, respectively. The obvious improvement of KS indicates that our model has better ability to discriminate between default and normal companies, which is critical to real applications.
- Machine learning methods achieve lower AUC and KS than graph learning methods. This is because traditional feature engineering seldom leverages interactive relations in corporate activities, yielding poor performance.
- Homogeneous GNNs obtain relatively better results than machine learning methods, with at least 0.5% increased AUC and 3.7% increased KS. The results verify the effectiveness of structural information in financial graphs. In addition, GAT performs better than GCN and GraphSAGE, implying that the attention-based aggregation is more efficient for distilling risky features.
- Heterogeneous GNNs are more effective than homogeneous GNNs, due to the capability of exploiting heterogeneous contents. However, these baseline HGNNs show the limited capability in taking full advantage of graph information

**Table 2: Comparison of performance results.**

| Method | SC21H1 | | SC21H2 | |
|--------|--------|--------|--------|--------|
| | AUC | KS | AUC | KS |
| LR | 0.7404 | 0.3266 | 0.7210 | 0.2803 |
| LightGBM | 0.7598 | 0.3570 | 0.7429 | 0.3102 |
| MLP | 0.7557 | 0.3565 | 0.7374 | 0.3086 |
| GCN | 0.7636 | 0.3703 | 0.7468 | 0.3245 |
| GraphSAGE | 0.7675 | 0.3814 | 0.7503 | 0.3346 |
| GAT | 0.7709 | 0.3854 | 0.7535 | 0.3477 |
| HAN | 0.7636 | 0.3782 | 0.7477 | 0.3376 |
| MAGNN | 0.7714 | 0.3848 | 0.7539 | 0.3518 |
| HGT | 0.7673 | 0.3801 | 0.7489 | 0.3454 |
| Simple-HGN | 0.7743 | 0.3978 | 0.7573 | 0.3681 |
| HetCAN$_{\backslash E}$ | 0.7810 | 0.4168 | 0.7672 | 0.3973 |
| HetCAN$_{\backslash A}$ | 0.7762 | 0.4035 | 0.7588 | 0.3790 |
| HetCAN$_{\backslash L}$ | 0.7796 | 0.4105 | 0.7647 | 0.3883 |
| HetCAN | **0.7877** | **0.4277** | **0.7781** | **0.4047** |

(i.e. holistic metapath context, multiple semantic connections) beneficial for assessing default risks, which results in suboptimal prediction.

## 4.3 Ablation Study

Next, we conduct ablation studies to validate the effects of each component in our model, which include three variants: HetCAN$_{\backslash E}$, HetCAN$_{\backslash A}$ and HetCAN$_{\backslash L}$. As shown in Table 2, the experimental results of all HetCAN variants in terms of AUC and KS deteriorate to some degree, which indicate that these components contribute to performance improvement. We further observe that almost all the variants work better than the baselines, which empirically demonstrates the effectiveness of leveraging these key graph properties for corporate default risk assessment. Additionally, it is worth noting that HetCAN$_{\backslash A}$ exhibits the most obvious performance degradation among all the variants, implying that our proposed attention scores play an essential role in taking advantage of collaborative metapaths to benefit the downstream prediction. Without considering collaborative metapaths, the model is limited to learn much effective risky features from the real financial graphs due to their massive noisy connections. Moreover, in spite of less performance loss compared with HetCAN$_{\backslash A}$, the results of HetCAN$_{\backslash L}$ confirm that with the guidance of feature learning by multiple semantic connections can further improve default prediction, which make the whole HetCAN obtain the best performance.

## 4.4 Co-attention Analysis

We make an in-depth analysis on the proposed co-attention mechanism by HetCAN and HetCAN$_{\backslash L}$. For each metapath, we compute the attention weights of the neighbors for all target nodes and then the average attention weights of each target node. Likewise, the neighbors are divided into two groups: single-metapath neighbors and multi-metapath neighbors. The lifting ratio of attention

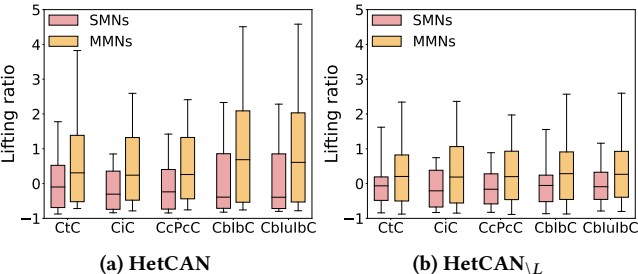

(a) HetCAN          (b) HetCAN$_{\backslash L}$

**Figure 4: Boxplot of the lifting ratio of attention weights for single-metapath neighbors and multi-metapath neighbors (SMNs and MMNs for short) on SC21H2 dataset (see SC21H1 results in Appendix A.5).**

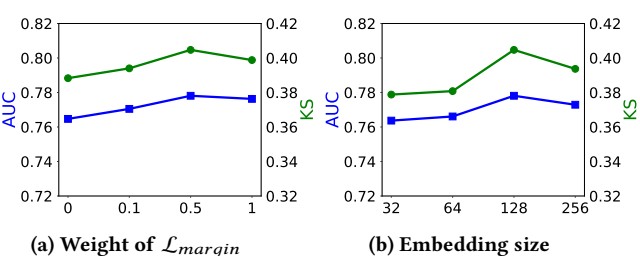

(a) Weight of $\mathcal{L}_{margin}$          (b) Embedding size

**Figure 5: Parameter study on SC21H2 dataset (see SC21H1 results in Appendix A.5).**

weights over average weights in each group are presented in Figure 4. The results show that the distribution of attention weights is relatively uniform between two groups. That is, multi-metapath neighbors have higher values than most single-metapath neighbors, implying that multiple semantic connections can be captured by our approach. Additionally, HetCAN further promotes the lifting ratio for multi-metapath neighbors in comparison to HetCAN$_{\backslash L}$, validating the effectiveness of pairwise importance learning for such connection patterns.

## 4.5 Parameter Sensitivity

We examine the parameter sensitivity of HetCAN through tuning a single parameter while keeping others unchanged. Figure 5 exhibits the performance results of different parameter settings on SC21H2 dataset. First, we study how the margin loss influences prediction by varying the weight between 0 and 1. In Figure 5a, the model performance continuously increases until reaching a saturation at about 0.5, indicating the effectiveness of constructing pairwise importance learning. But higher weights may cause large dominance of the margin loss in total objective function, which limits the learning effects. Second, we report the results of adjusting the embedding size in Figure 5b. With the increase of the embedding size, both AUC and KS rise and then drop. Small values hinder the model to learn sufficient information from data. Large values cause overfitting and redundancy, which may degenerate the performance. Here, the optimal size is set to around 128.

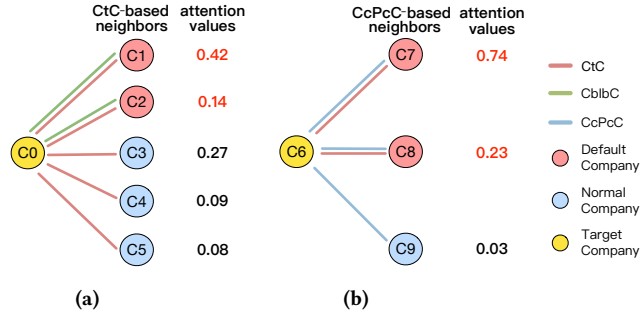

**Figure 6: Visualization of attention weights in HetCAN.**

## 4.6 Case Study

We adopt case studies on connection patterns to further investigate the advantages of HetCAN. Concretely, we pick two representative default companies that LightGBM (commonly used for the task) fails to identify, and visualize their attention weights in the co-attentive aggregation layers. In Figure 6a, we observe that C1 obtains the highest attention value for C0 under the metapath *CtC*, while C1 connects to C0 via another metapath *CbIbC*, implying that these two companies face the same industry risks. Our model can make use of the industrial relation (i.e. *CbIbC*) to alleviate noisy transactions. We provide another case that the collaborative metapaths can distill transitive risky features. In Figure 6b, C7 gets the highest attention value for C6 under *CcPcC*, due to their substantial fund transfers (i.e. *CtC*) between companies of the same controller, which is considered as high risk behaviors.

Additionally, we analyze the predictions by HetCAN. The subsets of companies which have the above two connection patterns (i.e. *CtC* and *CbIbC*, *CtC* and *CcPcC*) contain 16.59% and 1.89% of the total true positive predictions. HetCAN has detected 44.34% and 46.15% more default companies than LightGBM in these two subsets, respectively. Therefore, our HetCAN provides the interpretability and capability of taking advantage of multiple semantic connections for default prediction in the real financial graphs.

## 5 RELATED WORK

In this work, our focus is to use HGNN methods for the corporate default risk assessment problem.

**Heterogeneous Graph Neural Networks.** Many HGNNs extend the architectures from GNNs over homogeneous graphs [14, 22, 40], for capturing the structural and semantic information in heterogeneous graphs. These methods can be classified into two categories: (1) Metapath-based HGNNs adopt metapaths to aggregate neighborhood information of the same semantic and then fuse diverse semantics. HAN [44] proposed a graph attention network architecture to learn the importance of nodes and metapaths on metapath-based homogeneous graphs. MAGNN [11] extended HAN by exploiting intermediate nodes along metapath. (2) Metapath-free HGNNs directly aggregate information from neighbors by node and edge type aware modules to capture diverse semantics. HGT [18] employed a meta-relation based mutual attention mechanism for modeling heterogeneity. Simple-HGN [34] extended GAT

via learnable edge-type embeddings for attention calculation. To our knowledge, this is the first work which considers both massive noisy connections and multiple semantic connections in HGs. Moreover, our work leverages credit information rich in heterogeneous edges, which is crucial for financial scenarios.

**Co-Attention Mechanism.** Co-attention mechanism is derived from attention mechanism [1] by incorporating multiple inputs for joint learning. Recent works have verified its effect of capturing the interactions between different modalities [8, 31, 35]. In graph-learning communities, co-attention mechanism has been widely studied for applications: recommendation [16, 29], fake news detection [32] and traffic forecast [25]. Most studies adopt co-attention mechanism to model the influence between different aspects for better representation. Inspired by the idea, we take advantage of collaborative metapaths to distill the effective risky features in real financial graphs by designing a co-attention mechanism.

**Graph-based Risk Assessment.** Recent efforts attempt to exploit various corporate relations for risk assessment. Transaction networks contain key credit-related information and inter-company relationships, which are effective for risk analysis [23, 24]. Equity data form another considerable financial networks between companies and shareholders, beneficial for spotting credit risks [54]. A few works [2, 3] make use of investment relations and shared-news relations extracted from listed company information to analyze financial risks. Guarantee networks provide powerful transitive risky features for early warning [4], which are too sparse for small companies. Most closely related to ours is Yang's work on supply-chain mining for default prediction [48], where the upstream and downstream relationships are predicted by a semi-supervised model. Our industrial chain graph is focused on the concrete dependencies among industries and companies. In brief, our work differs from existing works in its setting of real banking graph data (i.e. fund, industry, equity), and the use of domain knowledge to reflect small company default risk from different perspectives.

## 6 CONCLUSION

In this paper, we focus on default risk assessment for small companies on heterogeneous graphs. By elaborately analyzing real financial graphs, we reveal the key graph properties related to the problem, which include massive noisy connections and multiple semantic connections. Based on these findings, we propose a novel HGNN model, named HetCAN, for corporate default risk assessment task. HetCAN aims to take advantage of collaborative metapaths to enhance the distillation of risky features by a co-attention mechanism, consisting of two attention scores and pairwise importance learning. Experiments on large-scale banking datasets verify that HetCAN is effective to default risk prediction for small companies. In future work, we will explore how to utilize more connections (like location data) and information (like economic data) and further improve the risk assessment on HGs.

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

# A APPENDIX

**Table 3: Data deficiency of small companies (SCs for short) and the effects of information supplement from view-specific neighbors.**

| Feature set | SCs | SCs filled by neighbors | | | |
|---|---|---|---|---|---|
| | | Equity | Industry | Fund | All |
| User profile | 18.4% | 17.3% | 15.1% | 7.1% | 7.0% |
| Credit status | 48.7% | 46.6% | 41.0% | 25.2% | 25.0% |
| Solvency | 89.5% | 88.9% | 83.8% | 58.1% | 58.0% |
| Operation | 55.3% | 54.4% | 48.9% | 22.4% | 22.3% |
| Activity | 7.1% | 6.0% | 4.7% | 1.6% | 1.4% |

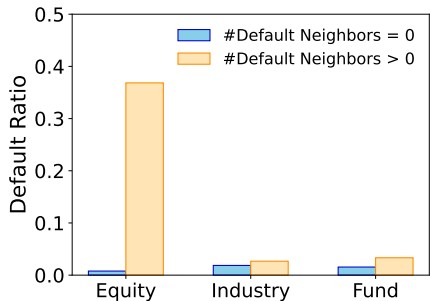

**Figure 7: The default ratio under different views.**

## A.1 Other Exploratory Analyses

Two statistical analyses are conducted to demonstrate two perspectives. First, small companies suffer from universal data deficiency, which can be alleviated by exploring their semantic connections. Second, the semantic connections have certain impact on small company default risk.

**Feature Analysis.** We investigate the status of banking data for expressing small companies. The investigation is performed on five sets of features, including user profile (i.e. client and account information), credit status (i.e. credit report and blacklist), solvency (i.e. contractual capability), operation (i.e. financial performance) and activity (i.e. account behavior and transaction action). For each set, we count the number of missing values over the total attributes for each small company and then compute the average missing rate. In Table 3, we find that small companies suffer significant deficiency in various aspects of attribute information.

Next, we verify that the data deficiency problem for small companies can be mitigated through aggregating their neighbors. For this purpose, we count missing attributes of a target company together with the metapath-based neighbors. To reveal the effects of different neighbors, five experimental groups are defined for comparison, including small companies themselves, small companies with neighbors guided by metapaths from a single view (equity, industry, fund), and small companies with neighbors from three views. As shown in Table 3, the results exhibit that feature sets

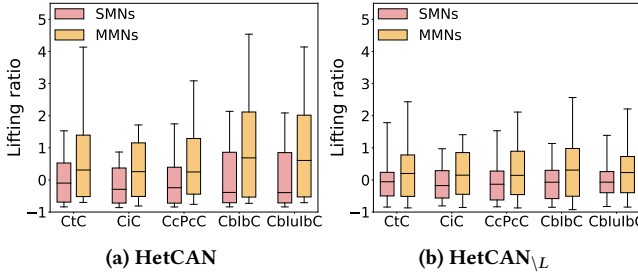

(a) HetCAN      (b) HetCAN$_{\setminus L}$

**Figure 8: Boxplot of the lifting ratio of attention weights for single-metapath neighbors and multi-metapath neighbors (SMNs and MMNs for short), which is conducted on SC21H1 dataset.**

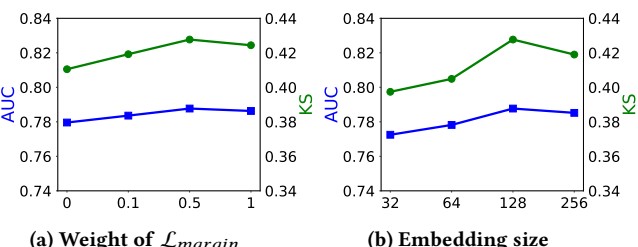

(a) Weight of $\mathcal{L}_{margin}$      (b) Embedding size

**Figure 9: Parameter sensitivity analysis on SC21H1 dataset.**

obtain enhancement to varying degree, where the average missing rates decrease by 80% at most. Besides, we observe that the semantic connections of each view improve the information of small companies and complete semantic connections achieve the best average missing rates.

**Impact analysis.** We further study the correlation between semantic connections and default risks. For each view, we collect the metapath-based neighbors of each small company. Then, we divide companies into two groups: companies with default neighbors and companies with no default neighbors. The default ratio (i.e. the proportion of default companies) is calculated in each group respectively. The comparative results are shown in Figure 7. We find that the semantic connections of three views have distinct impact on small company default risk, which motivates us to take advantage of corporate interactive relations for modeling the problem.

## A.2 Pseudo Code

Algorithm 1 shows the overall forward propagation process of our proposed HetCAN. We can stack multiple layers to expand the receptive field for better representation.

## A.3 Complexity Analysis

For simplicity, we use the number of nodes (denoted $|\mathcal{V}|$), the number of edges (denoted $|\mathcal{E}|$), the maximum size of the node attribute (denoted $d_1$), the maximum size of the edge attribute (denoted $d_2$), and the embedding size (denoted $d$). Given a metapath $\Phi$, we assume the number of metapath-based node pairs (denoted $n_{pair}^{\Phi}$), the number of metapath instances (denote $n_{path}^{\Phi}$), the length of metapath as $l_{\Phi}$. Then, the time complexity of feature

---

**Algorithm 1:** HetCAN forward propagation

**Input:** Heterogeneous graph $\mathcal{G} = (\mathcal{V}, \mathcal{E})$, node types $\mathcal{A}$, edge types $\mathcal{R}$, node attribute matrices $\{\mathbf{X}_A, \forall A \in \mathcal{A}\}$, edge attribute
     matrices $\{\mathbf{X}_R, \forall R \in \mathcal{R}\}$, target node set $\mathcal{U}$, metapath set $\mathcal{P}$, number of layers $L$

**Output:** Default probabilities $\hat{y}$ and global attention scores $g$

1 **for** *node type* $A \in \mathcal{A}$ **do**
2     Node feature transformation $\mathbf{h}_v^0 \leftarrow \mathbf{W}_A \mathbf{x}_v, \forall \mathbf{x}_v \in \mathbf{X}_A$;
3 **end**
4 **for** *edge type* $R \in \mathcal{R}$ **do**
5     Edge feature transformation $\mathbf{h}_e \leftarrow \mathbf{W}_R \mathbf{x}_e, \forall \mathbf{x}_e \in \mathbf{X}_R$;
6 **end**
7 **for** $u \in \mathcal{U}$ **do**
8     **for** $l = 1, \cdots, L$ **do**
9        **for** $\Phi \in \mathcal{P}$ **do**
10          Calculate metapath context embedding $[\mathbf{h}_{uv}^\Phi]^l$ for all $v \in \mathcal{N}_u^\Phi$ using Eq. (2) and Eq. (3);
11          Calculate local attention scores $[s_{uv}^\Phi]^l$ for all $v \in \mathcal{N}_u^\Phi$ using Eq. (4);
12        **end**
13        **for** $\Phi \in \mathcal{P}$ **do**
14          Calculate global attention scores $[g_{uv}^\Phi]^l$ for all $v \in \mathcal{N}_u^\Phi$ using Eq. (5);
15          Calculate attention values $[\alpha_{uv}^\Phi]^l$ for all $v \in \mathcal{N}_u^\Phi$ using Eq. (6);
16          Calculate the metapath-specific embedding $[\mathbf{h}_u^\Phi]^l$ using Eq. (7) and Eq. (8);
17        **end**
18        Calculate the attention weights $[\beta_\Phi]^l$ for all $\Phi \in \mathcal{P}$ using Eq. (12);
19        Fuse the metapath-specific embeddings $\mathbf{h}_u^l \leftarrow \sum_{\Phi \in \mathcal{P}} [\beta_\Phi]^l [\mathbf{h}_u^\Phi]^l$;
20     **end**
21     $\hat{y}_u \leftarrow \text{MLP}\left(\mathbf{h}_u^L\right)$;
22 **end**
23 **return** $\hat{y}, g$

---

transformation is $O(d_1 d |\mathcal{V}| + d_2 d |\mathcal{E}|)$. For metapath $\Phi$, the time complexity of the metapath context encoder is $O(l_\Phi d^2 n_{path}^\Phi)$, and the time complexity of the co-attentive aggregation is $O(d^2 |\mathcal{V}| + d n_{pair}^\Phi)$. Finally, the time complexity of HetCAN is $O(d_1 d |\mathcal{V}| + d_2 d |\mathcal{E}| + \sum_{\Phi \in \mathcal{P}} (l_\Phi d^2 n_{path}^\Phi + d^2 |\mathcal{V}| + d n_{pair}^\Phi))$. Despite the utilization of collaborative metapaths, we can see that the complexity of the co-attentive aggregation is on par with the classic metpath-based HGNNs [44].

## A.4 Baseline Settings

We report the detailed settings of the baseline models in our experiments. For LightGBM, the tree number is set as 1000 and the tree depth is set as 7, respectively. For MLP, we have $L = 2, d = 128$. For homogeneous GNNs, the common hyperparameters include $L = 3$, $d = 128$, $\sigma = \text{ReLU}(\cdot)$. We adopt GCN as the aggregator function for GraphSAGE, and $n_h = 1$ for GAT. For metapath-based HGNNs (i.e. HAN and MAGNN), we have $L = 2, d = 128, \sigma = \text{ReLU}(\cdot)$, $n_h = 1$. For HGT, we use layer normalization in each layer, and $L = 3, d = 64, n_h = 8$. For Simple-HGN, we have $L = 3, d = d_e = 128$, $\sigma = \text{ELU}(\cdot), n_h = 8, \beta = 0.05$. Additionally, for all GNN methods, we randomly initialize the model parameters with a Xavier initializer and choose Adam as the optimizer. Moreover, we respectively set the batch size to 256, the learning rate to 0.001, the sampling

size of neighbors to 10, the dropout rate to 0.5 and the weight decay to 0.01.

## A.5 Other Experimental Results

The experiment results on SC21H1 dataset, which include the co-attention analysis shown in Figure 8 and the parameter analysis shown in Figure 9, respectively. We can have similar conclusions in the former evaluation works.

