# OpenReview forum: "Collaborative Metapath Enhanced Corporate Default Risk Assessment on Heterogeneous Graph"
_ACM.org/TheWebConf/2024/Conference — TheWebConf24_

### Official Review · Reviewer_bX6S · 2023-11-19

**Novelty:** 4
**Technical Quality:** 4

**Review:**

The work presents a significant advancement in utilizing HGNNs for risk assessment, offering both innovative methodology and promising empirical results. However, further exploration into the model's broader applicability and practical implementation challenges would enhance its contribution to the field.

**Questions:**

1、I am having difficulty understanding the proportion between companies and users in the two datasets mentioned in the manuscript. For instance, the dataset lists 12,793,669 companies and 12,014,090 individuals. Does the dataset imply that there are more companies than individuals on average?
2、The manuscript highlights 'Noisy connection analysis' as a focal point and innovation. However, I did not find a clear definition of 'noisy' in the real-world scenarios. Could you please specify what types of realities or phenomena are referred to as 'noisy' in your research? How do you demonstrate the effectiveness of eliminating such noise? An explanation grounded in practical significance would be of particular interest to me. (Maybe the author explained it somewhere in the article, but since it is a contribution and a highlight, it is really confusing that it cannot appear in a prominent place.)
3、The manuscript appears to utilize only two datasets for the study.
4、I noticed that the comparative methods used in the study primarily include basic machine learning models, with limited direct comparison to the SOTA in the 'Corporate default risk assessment' field. It would be beneficial to understand how these basic models align with the specific context of corporate default risk, and whether including more directly relevant SOTA comparisons could provide a more robust evaluation of your proposed methods.
5、There are too many symbols, which is a bit difficult to understand and has low readability.

**Reviewer Confidence:**

3: The reviewer is confident but not certain that the evaluation is correct

**Scope:**

3: The work is somewhat relevant to the Web and to the track, and is of narrow interest to a sub-community

---

### Official Review · Reviewer_V812 · 2023-11-20

**Novelty:** 5
**Technical Quality:** 5

**Review:**

The paper studies the default risk assessment for small companies, which is a tough problem in financial services.

Strong points:
+ The paper identifies two challenges for the default risk assessment through extensive analysis in the real-world datasets: noisy connection and multiple semantic connection.

+ Motivated by the unique challenges, the paper improves current HGNNs by introducing co-attention aggregation and pairwise importance learning. The motivation is reasonable and the proposed solution seems sound.

+ The paper conductive extensive experiments as well as ablation studies on two industrial datasets, and the experimental results show that the proposed model works well and each well-designed component is important.

Weak points:
- “Existing HGNNs [11, 44] discard intermediate nodes and edges along metapath, which results in significant loss of information.”(line 340~342). The model in paper [11,55] have consider the intermediate nodes and edges along metapath. And paper [55] encodes  metapath instance with LSTM, paper [16] encodes metapath instance with CNN, while this paper encode metapath with GRU, what's the difference or strengths of the encoding way.

- How the proposed method address the noisy connection issue is not clear. Why can the global attention filter valuable/noisy signals? "The node-level attention is extended from classic GAT, which suffers from the limitation of discerning valuable/noisy signals."(line 445~447), why ?

-  The noisy connection and multiple semantic connection issues are also existed in public datasets. I think adding more experimental results on more public datasets with more metrics (Micro/Macro F1) could make the conclusion more convincing.

**Questions:**

Please refer to the main comments.

**Reviewer Confidence:**

3: The reviewer is confident but not certain that the evaluation is correct

**Scope:**

3: The work is somewhat relevant to the Web and to the track, and is of narrow interest to a sub-community

---

### Official Review · Reviewer_YV6s · 2023-11-22

**Novelty:** 5
**Technical Quality:** 4

**Review:**

This paper proposes a heterogeneous graph co-attention network to achieve corporate default risk prediction for small companies, which are weak when facing financial risks. The authors point out two main challenges of current methods, massive noises in real-world financial graphs and transitive default risk. To solve these two issues, this paper designs two methods respectively. Firstly, to discern noises, authors design a co-attention mechanism based on collaborative metapaths. Then, they apply a pairwise importance learning to model multiple semantic connections, thereby distilling transitive risky features. The main contribution of this paper is considering multiple semantic connections in financial graphs based on metapaths, reducing the effects of noises.

A few issues:

1. The authors claim that “Multiple semantic connections increase transitive default risk.” Is there any theoretical basis or empirical evidence? In section 2.2, authors mention that companies with default neighbors have higher default risk, and more metapaths between one company and its default neighbors lead to higher default risk. Is this the definition of “transitive risky features”? How to prove the transitive risk is influenced by “multiple semantic”? Meanwhile, the description in Introduction section is quite confusing, authors may reorganize it.

2. The definition of metapath is not clear. In page 3 (Definition 2.2), authors give the definition of metapath. However, they didn’t explain how to extract a metapath from a financial graph. For example, in Figure 1(b), “CtC”is a metapath, why? Can “CtCcP”be a metapath?

3. In section 3.2.1, authors apply the local attention score to obtain the metapath context embedding. But how do you measure the similar features of neighbors? How do you extract holistic metapath context? Can you provide any case study or ablation study to analyze the effect of different scales of holistic metapath context?

4. Please check the typos carefully. For example, in Definition 2.2, the notation of metatpath instance is inconsistent.

**Questions:**

1. How do you define “metapath” in financial graphs?

2. How do you explore transitive risky features? For one node (company), how to extract its transitive risky features?

**Reviewer Confidence:**

3: The reviewer is confident but not certain that the evaluation is correct

**Scope:**

3: The work is somewhat relevant to the Web and to the track, and is of narrow interest to a sub-community

---

### Official Review · Reviewer_RMLP · 2023-11-24

**Novelty:** 5
**Technical Quality:** 5

**Review:**

The paper contributes by conducting quantitative analyses on real financial graphs, proposing the Heterogeneous Graph Co-
Attention Network (HetCAN) model for corporate default risk assessment on heterogeneous graphs, incorporating collaborative metapaths and a co-attention mechanism. The thorough evaluation on large-scale banking datasets demonstrates the superiority of HetCAN over other models, especially in predicting default risks for small companies. Additionally, the open-sourcing of the code promotes transparency and collaboration in the research community.
In my opinion, the strongest part of the paper is the case study, that demonstrates that HetCAN effectively utilizes connection patterns, including collaborative metapaths, to improve default prediction in real financial graphs. The model's interpretability is emphasized through visualizations of attention weights, and its performance surpasses that of the commonly used LightGBM in identifying default companies with specific connection patterns.
The weakest point is the technical difficulty of the paper. Although the authors do their best with very good illustrative examples, sound formalization, and an accurate (to my knowledge) review of the state of the art, I found that the many different methodological layers can make the applicability of this approach to other domains quite limited.

**Questions:**

- Do you think that your approach can be used in other domains? Is it generalizable?

**Reviewer Confidence:**

2: The reviewer is willing to defend the evaluation, but it is likely that the reviewer did not understand parts of the paper

**Scope:**

2: The connection to the Web is incidental, e.g., use of Web data or API

---

### Official Review · Reviewer_NDkf · 2023-11-29

**Novelty:** 5
**Technical Quality:** 6

**Review:**

Strongness:
1. The paper proposes a novel approach called HetCAN that utilizes collaborative metapaths to effectively assess corporate default risks.
2. The paper provides a comprehensive evaluation of the proposed approach on real-world financial graph data.
3. The paper demonstrates the effectiveness of leveraging key graph properties for modeling and risk assessment.
4. The paper includes an ablation study to validate the effects of each component in the proposed model.

Cons:
1. The paper assumes that the graph data is complete and accurate, which may not always be the case in real-world scenarios.
2. The paper does not provide a detailed comparison with other state-of-the-art approaches for corporate default risk assessment.
3. The paper does not discuss the limitations and potential drawbacks of the proposed approach.

**Questions:**

1. Can you provide more details on the limitations of the proposed approach and how they could be addressed in future work?
2. How does the proposed approach compare to other state-of-the-art methods for corporate default risk assessment, and what are the key differences?
3. Can you discuss the scalability of the proposed approach and how it could be applied to even larger financial graphs?
4. How sensitive is the proposed approach to the choice of hyperparameters, and how did you select the optimal values for the experiments?
5. Can you provide more insights into the interpretability of the proposed approach and how it could be used to identify key risk factors for small businesses?
6. How does the proposed approach handle missing or incomplete data in the financial graph, and what are the potential implications for risk assessment accuracy?
7. Can you discuss the potential ethical implications of using graph-based approaches for corporate default risk assessment, and how they could be addressed in practice?

**Reviewer Confidence:**

3: The reviewer is confident but not certain that the evaluation is correct

**Scope:**

3: The work is somewhat relevant to the Web and to the track, and is of narrow interest to a sub-community

---

### Decision · Program_Chairs · 2024-01-22

**Decision:**

Accept

**Comment:**

This paper proposed novel Heterogeneous Graph Co-Attention Network (HetCAN) for corporate default risk assessment. HetCAN aims to take advantage of collaborative metapaths to distill effective risky features by a co-attentive aggregation mechanism, consisting of two attention scores and pairwise importance learning. First, the local attention score models the importance of neighbors under each metapath by considering holistic metapath context. Second, the global attention score further adjusts the importance of neighbors by combining these local attention scores to filter valuable/noisy signals. Then, HetCAN employs pairwise importance learning to enhance attention scores of multi-metapath neighbors for risky feature distillation.

 The reviewers appreciate the novelty and experimental evaluation. The authors have addresssed most of the comments of the reviewers as well.